# Retinopathy among women with hypertensive disorders of pregnancy attending hospitals in Mbarara city, south-western Uganda: a cross-sectional study

Ibrahimu Zamaladi,[1] Sam Ruvuma,[1] Carmel M Mceniery,[2] Teddy Kwaga,[1] Ian B Wilkinson,[2] Daniel Atwine,[3,4] Henry Mark Lugobe [ID] [2,5]

For numbered affiliations see end of article.

**Correspondence to**
Henry Mark Lugobe;
henrylugobe@must.ac.ug

## ABSTRACT

**Objective** Retinopathy is one of the complications occurring among women with hypertensive disorders of pregnancy. We sought to determine the prevalence and factors associated with retinopathy among women with hypertensive disorders of pregnancy in southwestern Uganda.

**Design** This was a hospital-based cross-sectional study from November 2019 to March 2020.

**Setting** Three selected hospitals in Mbarara city, south-western Uganda.

**Participants** The study included all pregnant women with hypertensive disorders of pregnancy.

**Primary and secondary outcome measures** The participants were screened for retinopathy using a fundus camera. Data on participant's sociodemographics, obstetrics and medical factors were collected. The prevalence of retinopathy was determined and multivariable logistic regression was used to determine the independent factors associated with retinopathy.

**Results** A total of 216 women with hypertensive disorders of pregnancy were enrolled in this study. The prevalence of retinopathy was 60.2% (130/216). The most common retinal lesions were grade 1 retinopathy (narrowing of arterioles) accounting for 86.9% (113/130), grade 3 (retinal haemorrhages) was present in 10% (13/130) of women and grade 4 (papilloedema) in 3% (4/130). In an adjusted analysis, severe hypertension was significantly associated with retinopathy (aOR=2.8; 95% CI: 1.36 to 5.68). Grandmultigravida women were also associated with retinopathy (aOR=2.4; 95% CI: 0.99 to 5.72) with a tendency towards significance, p=0.051.

**Conclusions** In our study, retinopathy was common among women with hypertensive disorders of pregnancy. Women presenting with severe hypertension were likely to have retinopathy. There is a need to integrate screening for retinopathy in the care cascade of women with hypertensive disorders of pregnancy.

## STRENGTHS AND LIMITATIONS OF THIS STUDY

⇒ A portable non-mydriatic fundus camera was used to examine for retinopathy.
⇒ Blood pressure was only done at one time point.
⇒ Routine screening of participants for diabetes or glucose intolerance during their pregnancy was not done.
⇒ Biochemical tests to determine presence of hypertension with end organ damage were not done.

may worsen or alter during pregnancy as a result of haematologic and metabolic change.[1] During pregnancy, elevated blood pressure poses various risks including placental abruption, intrauterine growth retardation, premature delivery, future cardiovascular diseases and injury to other organs including the eye.[2] Ocular involvement is common in a majority of cases of hypertensive disorders of pregnancy. Common symptoms are blurry vision, photopsia, scotomas and diplopia.[3] Visual disturbances and retinopathy in hypertensive disorders of pregnancy seem to be a frequent phenomenon.[4] The global prevalence of hypertensive disorders of pregnancy ranges between 5% and 10% and fundus changes are seen in 40%–100% of these patients.[5] In Africa, hospital-based prevalence of hypertensive disorders of pregnancy is 8% in Ethiopia[6] and 17% in Nigeria.[7] In Uganda, hypertensive disorders of pregnancy accounted for 12% of maternal deaths in four referral hospitals,[8] while at Mbarara regional referral hospital they accounted for 22% of the maternal deaths.[9]

Studies in Malaysia and India showed that 40% and 59% of women with hypertensive disorders of pregnancy, respectively, presented with retinal changes while in Nepal

## INTRODUCTION

Retinopathy in pregnancy is a term that defines the retinal pathologies seen uniquely in pregnancy or more commonly in conditions that

it was 14%.[10] In Uganda, a study done at Mulago National Hospital showed a prevalence of 16.5% of retinal changes in pre-eclampsia/eclampsia patients.[11] Retinopathy among women with hypertensive disorders of pregnancy is associated with fetal birth weight, serum uric acid, proteinuria, severe hypertension and maternal age.[12–15]

However, there is a paucity of information on retinopathy among women with hypertensive disorders of pregnancy in sub-Saharan Africa where hypertension in pregnancy is associated with significant maternal morbidity and mortality. We sought to determine the prevalence and factors associated with retinopathy among women with hypertensive disorders of pregnancy in south-western Uganda.

## MATERIALS AND METHODS

### Study design and setting
We conducted a hospital-based, cross-sectional study of women with hypertensive disorders of pregnancy, at three selected hospitals in south-western Uganda from November 2019 to March 2020. These hospitals were Mbarara Regional Referral Hospital, Mayanja Memorial Hospital and Divine Mercy Hospital. Mbarara regional referral hospital is a government-owned referral hospital that offers free services as well as a teaching hospital for the medical school of Mbarara University of Science and Technology. The hospital conducts about 10 000 deliveries every year. Mayanja Memorial Hospital is a private hospital with a capacity of 60 beds, while Divine Mercy Hospital is also a private hospital with a capacity of 22 beds. All these facilities offer antenatal, labour, delivery and postnatal care services. These facilities receive referrals from 16 districts in south-western Uganda.

### Study participants
Our study population included all pregnant women at ≥20 weeks of gestation with hypertensive disorders of pregnancy who were receiving care either as inpatients at the antenatal ward or as outpatients at the antenatal clinic. The blood pressure was measured at admission or during the antenatal clinic visit. Hypertension was defined as systolic blood pressure ≥140 mm Hg or diastolic blood pressure ≥90 mm Hg. Women with ocular diseases that prevented adequate fundus examination, such as cataracts, corneal opacities and ocular trauma, were excluded from the study. All mothers presenting at the maternity ward or antenatal clinic had their blood pressure checked as part of routine clinical care and eligible participants were consecutively enrolled in the study.

### Variables and data sources
Hypertensive disorders of pregnancy were classified into eclampsia, pre-eclampsia, gestational hypertension, chronic hypertension and chronic hypertension with superimposed pre-eclampsia, according to consensus definitions.[16] All pregnant women who had hypertensive disorders of pregnancy either diagnosed as inpatient or outpatient were screened for retinopathy. The primary outcome for our study was retinopathy and was determined using Keith and Wagner (KW) classification.[17]

Grade 1: Mild generalised arterial attenuation, particularly of small branches, with broadening of the arteriolar light reflex.

Grade 2: marked generalised narrowing and focal attenuation of arterioles associated with arteriovenous nipping/narrowing (Salus' sign).

Grade 3: grade 2 changes plus copper-wiring of arterioles, banking of veins distal to arteriovenous crossings (Bonnet sign), tapering of veins on either side of the crossings (Gunn sign). Flame-shaped haemorrhages, cotton wool spots, and hard exudates are also present.

Grade 4: all changes of grade 3 plus silver wiring of arterioles and papilloedema.

Uncorrected visual acuity was measured using a Snellen chart and E-chart. Anterior segment examination was done using a torch, while funduscopy was done using both a direct ophthalmoscope (brand Keeler) after dilating the pupil with tropicamide 1%, and used a portable non-mydriatic fundus camera (Forus 3nethra non-mydriatic, Forus Health, Bengaluru, India). Fundus changes were recorded and all fundus photos were interpreted by a consultant retinal specialist of Mbarara University Referral Hospital Eye Centre. The fundus pictures were graded for quality before KW grading.

The independent variables included sociodemographic characteristics (age, level of education, address, district of origin and occupation), medical factors (known history of diabetes mellitus, hypertension) and obstetric factors (gravidity, severity of hypertension (systolic blood pressure≥160 mm Hg and or diastolic blood pressure≥110 mm Hg),and gestational age). At enrolment into the study, participants were interviewed and information was obtained using an interviewer-administered questionnaire.

### Sample size calculation
We calculated a total sample size of 216 participants using the formula for cross-sectional sample size calculations of Kish Leslie with the following assumptions.

$n = (Z\alpha^2 p (1 p))/\delta^2$

where

n=sample size estimate.

p=prevalence (16.5 %) as estimated by a study done at Mulago hospital[11]

δ=acceptable margin of error 5% (0.05).

Zα= standard normal deviation at 95% CI: 1.96.

1–p=the probability of having retinopathy (1–0.17) = 0.83

$n = (1.96^2 \times 0.17 (1–0.17)) /0.05^2$.

n=216 participants

### Statistical analysis
Data were entered into Microsoft excel 2010 and were imported into STATA V.15.0 software (College Station, Texas, USA) for analysis. Descriptive statistics were

presented as means and SD for continuous variables and proportions for categorical variables. the proportion of patients with retinopathy was calculated as the number of participants with retinal changes over the total number of study participants and expressed as a percentage with its corresponding 95% CI. Retinal changes were classified according to KW classification into grade 1, grade 2, grade 3 and grade 4, and the proportion of each grade was presented in a frequency table. Bivariate analysis using Pearson $\chi^2$ test and logistic regression analysis was performed comparing participant factors with retinopathy. At bivariate analysis, variables with statistically significant association with retinopathy (p<0.05), those with p<0.1 and those with biological plausibility with retinopathy, for example, age were included in the multivariable logistic regression model. A manual Pearson backward elimination was used to build the final model. The independent factors associated with retinopathy in the final multivariable model were reported together with their adjusted ORs and 95% CI. The level of significance was p<0.05.

### Patient and public involvement

Patients and or the public were not involved in the design or conduct or reporting or dissemination plans of this research.

## RESULTS

### Study profile

A total of 5008 pregnant women attended Mbarara Regional Referral Hospital, Mayanja Memorial Hospital and Divine Mercy Hospital during the study period. Of these, 216 women with hypertensive disorders of pregnancy were consented and enrolled in the study. A total of 208 (96.3%), 6 (2.8%) and 2 (0.9%) were recruited from Mbarara Regional Referral Hospital, Mayanja Hospital and Divine Mercy Hospital, respectively. Of the participants, 88.4% (191/216) had pre-eclampsia, 10.2% (22/216) had eclampsia and 1.4% (3/216) had gestational hypertension.

### Participants' characteristics

Table 1 shows the sociodemographic and medical characteristics of women enrolled in the study. Participants had a mean age of 27.6±6.37 years with majority aged less than 35 years (81.7%), Ugandan (90.3%) and either unemployed or peasants (74.5%). Participants predominantly had 37 or more gestational weeks (70.4%) and were either multigravida or grand multigravida mothers (72%). One in four had severe hypertension (26.4%) at the time of the study. Majority of participants (214/216) had good uncorrected visual acuity of 6/6–6/18, and only 1/216 had severe visual impairment. A third of the participants presented with blurry vision. There was no case of complete blindness in our study.

### Prevalence of hypertensive retinopathy among pregnant women

Of the 216 women with hypertensive disorders of pregnancy attending hospitals in Mbarara city and enrolled

**Table 1** Baseline characteristics of participants

| Characteristics | | N (%) |
|---|---|---|
| Age in years, mean (SD) | | 27.6 (6.37) |
| Age category | <25 | 77 (35.4) |
| | 25–34 | 100 (46.3) |
| | 35–46 | 39 (18.1) |
| Nationality | Ugandan | 195 (90.3) |
| | Non-Ugandan | 21 (9.7) |
| Occupation | Unemployed | 84 (38.9) |
| | Peasant | 77 (35.6) |
| | Business | 30 (13.9) |
| | Professional | 17 (7.9) |
| | Labourer | 8 (3.7) |
| Known history of diabetes mellitus | No | 209 (96.8) |
| | Yes | 7 (3.2) |
| Severe hypertension | No | 159 (73.6) |
| | Yes | 57 (26.4) |
| Gravidity | Primigravidas | 61 (28.2) |
| | Multigravida | 111 (51.4) |
| | Grand multigravida | 44 (20.4) |
| Gestational age (weeks) | <37 | 64 (29.6) |
| | ≥37 | 152 (70.4) |
| Ocular symptoms | No complaints | 42 (19.4) |
| | Blurry vision | 71 (32.9) |
| | Reduced vision | 7 (3.3) |
| | Photopsia | 2 (0.9) |
| | Inability to focus | 1 (0.5) |
| | Visual field defect | 1 (0.5) |
| Visual acuity | 6/6–6/18 | 214 (99.0) |
| | <6/18–6/60 | 1 (0.5) |
| | 6/60–3/60 | 1 (0.5) |

in the study, 130 had retinopathy, giving a prevalence of 60.2% (95% CI: 53.5 to 66.5). No age-specific disparities where noted with regard to prevalence of retinopathy, p=0.136 as shown in table 2.

Of the 130 women with retinopathy, 113 (86.9%) presented with arteriolar narrowing (figure 1). Severe retinopathy characterised by retinal haemorrhages and papilloedema (figure 2) was also recorded. No one had grade 2 retinopathy and no retinal detachment as shown in table 3.

### Factors associated with retinopathy among women with hypertensive disorders of pregnancy

Table 4 shows results of bivariate and multivariate analysis. In bivariate analysis, the only factors that were significantly associated with retinopathy were gravidity (p=0.049) and severe hypertension (p=0.005).

In multivariate analysis, severe hypertension was the only factor significantly associated with retinopathy after adjusting for age, gravidity and gestational age, p=0.005.

**Table 2** Prevalence of hypertensive retinopathy among pregnant women

| Prevalence type | N | N | % (95% CI) | P value |
|---|---|---|---|---|
| Overall | 216 | 130 | 60.2 (53.5 to 66.5) | NA |
| Age-specific | 77 | 44 | 57.1 (45.7 to 67.9) | 0.136 |
| <25 | 100 | 57 | 57.0 (47.0 to 66.5) | |
| 25–34 | 39 | 29 | 74.4 (57.8 to 86.0) | |
| 35–46 | | | | |

NA, not applicable.

The odds of having retinopathy were 2.8 times higher among women with severe hypertension as compared with those without, OR=2.8; 95% CI: 1.36 to 5.68. Also, the odds of having retinopathy were 2.4 times higher among grandmultigravida women as compared with those with multigravida women, OR=2.4; 95% CI: 0.99 to 5.72, although this did not attain statistical significance, p=0.051.

## DISCUSSION

In our study, the overall prevalence of retinopathy among women with hypertensive disorders of pregnancy was 60.2% with most of the lesions being grade 1. Severe hypertension was independently associated with retinopathy.

The prevalence in our study is comparable with what was found in other studies done in Pakistan and Malaysia, where the prevalence of retinopathy among women with hypertensive disorders of pregnancy was 51.9%–59%.[17 18] Our result is lower than what was found in Kenya (90.8%).[19] The difference may be because the study in Kenya considered postpartum women who had pre-eclampsia with severe features. Other studies have shown a lower prevalence of 12%–16.5% compared with what was found in our study.[11 20 21] The difference could be because in these studies a direct ophthalmoscope was the diagnostic tool used for retinopathy which could have possibly underestimated the participants with early retinal lesions on funduscopy compared with our study where a fundus camera was used.

From our results, grade 1 retinopathy characterised by generalised arteriolar narrowing was the the most common retinal lesion, while grades 3 and 4 retinopathy were the least observed among women with hypertensive disorders of pregnancy. This is similar to other studies, where grade 1 retinopathy was the the most common form

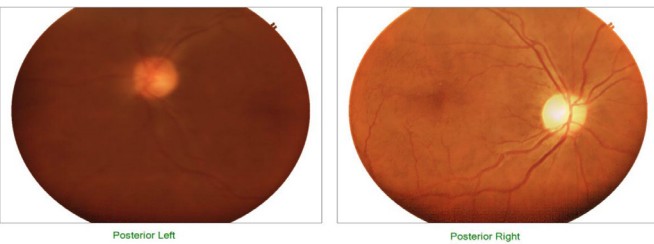

**Figure 1** Hypertensive retinopathy grade 1: narrowing of the arterioles in both eyes.

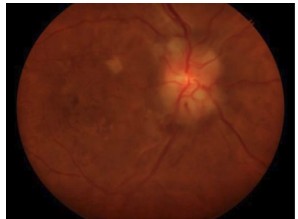
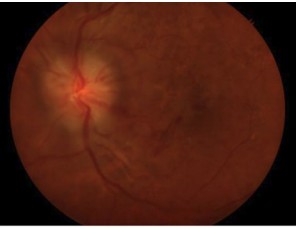

Right eye Left eye

**Figure 2** Hypertensive retinopathy grade 4 in both eyes: exudates, flame-shaped haemorrhages and papilloedema.

of retinopathy and grades 3 and 4 retinopathy were the least common among women with hypertensive disorders of pregnancy.[17 18 22] However, a cross-sectional study done at Mulago National Hospital in Uganda among women with pre-eclampsia/eclampsia found that the the most common patterns of ocular manifestations were optic disc oedema 19 (26.7%), retinal haemorrhages 17 (22.9%) and cotton wool spots 16 (21.6%).[11] This high proportion of severe retinal lesions compared with that observed in our study may be explained by the use of a direct ophthalmoscope with a possibility of underestimating early retinal lesions on funduscopy, while in our study, a fundus camera was used with the ability to detect retinal lesions even at an early stage. Compared with our study, this study considered women who were presenting for maternity admission and these usually have severe disease, while our study also considered women presenting in the outpatient antenatal clinics. Grade 1 retinal changes maybe due to constriction of retinal arterioles following elevated blood pressure.[15] Retinal changes seen in grades 3 and 4 could be due to endothelial damage, systemic inflammation, ischaemia caused by hypoperfusion, or retinal oedema caused by hyperfusion.[23] Presence of hypertensive retinopathy in pregnant women with hypertensive disorders of pregnancy could be an indicator of vascular changes in placental circulation[11] and the severity of maternal fundus changes may indirectly indicate the status of the placental vasculature.[15]

In our study, severe hypertension was associated with retinopathy among women with hypertensive disorders of pregnancy. This finding is similar to what was reported in others studies.[14 15 18 21 24] The retinal changes observed

**Table 3** Description of retinal lesions found among women with hypertensive disorders of pregnancy according to Keith and Wagner classification

| Retinal lesions (n=130) | | N (%) |
|---|---|---|
| Grade 1 | Arteriolar narrowing | 113 (86.92) |
| Grade 3 | Dot and blot haemorrhages | 6 (4.62) |
| | Flame-shaped haemorrhages | 3 (2.30) |
| | Cotton wool spots | 2 (1.54) |
| | Exudates | 2 (1.54) |
| Grade 4 | Papilloedema | 2 (1.54) |
| | Maculoedema | 2 (1.54) |

**Table 4** Factors associated with retinopathy among women with hypertensive disorders of pregnancy

| Variable | Categories | Retinopathy N (%) | Crude OR (95% CI) | P value | Adjusted OR (95% CI) | P value |
|---|---|---|---|---|---|---|
| Age | <25 | 44 (57.1) | Ref. | 0.124 | Ref. | |
| | 25–34 | 57 (57.0) | 0.9 (0.55 to 1.81) | | 0.9 (0.42 to 1.72) | 0.655 |
| | 35–46 | 29 (74.4) | 2.2 (0.93 to 5.08) | | 1.5 (0.54 to 4.18) | 0.435 |
| Nationality | Ugandan | 116 (59.5) | Ref. | 0.519 | | |
| | Non-ugandan | 14 (66.7) | 1.4 (0.53 to 3.53) | | | |
| Occupation | Unemployed | 56 (66.7) | 1.8 (0.62 to 5.11) | 0.4723 | | |
| | Peasant | 41 (53.3) | 1.0 (0.35 to 2.90) | | | |
| | Business | 19 (63.3) | 1.5 (0.46 to 5.14) | | | |
| | Professional | 9 (52.9) | Ref. | | | |
| | Labourer | 5 (62.5) | 1.5 (0.27 to 8.27) | | | |
| Severe hypertension | No | 87 (54.7) | Ref. | 0.005 | Ref. | |
| | Yes | 43 (75.4) | 2.5 (1.29 to 5.01) | | 2.8 (1.36 to 5.68) | 0.005 |
| Gravidity | Multigravid | 60 (54.1) | Ref. | 0.049 | Ref. | |
| | Primigravida | 37 (60.7) | 1.3 (0.69 to 2.47) | | 1.2 (0.61 to 2.54) | 0.556 |
| | Grandmultigravida | 33 (75.0) | 2.5 (1.17 to 5.55) | | 2.4 (0.99 to 5.72) | 0.051 |
| Gestational age | ≥37 | 88 (57.9) | Ref. | 0.287 | Ref. | |
| | <37 | 42 (65.6) | 1.4 (0.76 to 2.55) | | 1.1 (0.60 to 2.18) | 0.681 |
| Diabetes mellitus | No | 123 (58.9) | Ref. | NA | | |
| | Yes | 7 (100) | 1 (NA) | | | |

NA, not applicable.

may be due to constriction of the retinal arterioles which occurs following the elevation of blood pressure.[15] The elevation in blood pressure involves a spectrum of retinal microvascular signs. These signs typically include retinal arteriolar narrowing, arteriovenous nicking retinal haemorrhages, microaneurysms, and in severe cases, optic disc and macular oedema.[25]

Grandmultigravidity was associated with retinopathy although the association was not statistically significant. We have not found any previous studies that have demonstrated an association between grandmultigravidity and retinopathy.

Our study had some limitations. First, we only identified women who reported a known history of diabetes mellitus. However, we were not able to routinely screen the participants for diabetes or glucose intolerance during their pregnancy. Second, biochemical tests that determine hypertension with end organ damage were not done as we relied only on the blood pressure. Therefore, we were not able to determine whether hypertension with end organ damage is a risk factor. Third, blood pressure was only done at one time point and therefore we could have missed out on participants who developed hypertension later in their pregnancy. Fourth, since we considered all pregnant women who presented with hypertension, we did not highlight the role of chronic hypertension in this study.

## CONCLUSIONS

From our study, retinopathy among women with hypertensive disorders of pregnancy is common. Funduscopy should form part of the routine examination for all women with hypertensive disorders of pregnancy to identify women with retinal lesions early enough and have them linked to care at the ophthalmology clinic for proper eye care management. Attention should also be given to women with severe hypertension and grandmultigravidas as these are more likely to have retinal lesions. There is a need for further research to document the long-term outcomes of women with retinopathy and hypertensive disorders of pregnancy as regards to their long-term vision.

**Author affiliations**
[1]Department of Ophthalmology, Mbarara University of Science and Technology, Mbarara, Uganda
[2]Division of Experimental Medicine and Immunotherapeutics, Department of Medicine, University of Cambridge, Cambridge, UK
[3]Department of Clinical Research, Soar Research Foundation, Mbarara, Uganda
[4]Department of Community Health, Mbarara University of Science and Technology, Mbarara, Uganda
[5]Department of Obstetrics and Gynecology, Mbarara University of Science and Technology, Mbarara, Uganda

**Acknowledgements** We would like to convey our appreciation to the study participants for accepting to be part of this study, Ms. Naiga Patience and Ms. Florida Tusiimiraho our research assistants, Soar Research Foundation staff for

support in data management and to all the hospital administrators who offered permission to conduct this study in their hospitals.

**Contributors** IZ, SR, CMM, TK, IBW, DA and HML conceived the study. IZ, CMM, TK, IBW and SR developed the data collection. DA, IZ and HML conducted the analysis. HML drafted the manuscript. All authors reviewed and approved the final manuscript. HML is the guarantor.

**Funding** Research reported in this publication was supported by the Forgarty International Center of the National Institute of Health under award number D43TW011401 to HML. The content is solely the responsibility of the authors and does not necessarily represent the official views of the NIH.The research reported here was supported by the Commonwealth Scholarship Commission and the Foreign, Commonwealth and Development Office in the UK to HML. The author is grateful for their support. All views expressed here are those of the author(s) not the funding body. HML is funded by the Commonwealth Trust. This research was supported, in part, by the NIHR Cambridge Biomedical Research Centre (BRC-1215-20014) to CMM. The views expressed are those of the authors and not necessarily those of the NIHR or the Department of Health and Social Care. The funders had no role in study design, data collection, and analysis, decision to publish or preparation of the manuscript.

**Competing interests** None declared.

**Patient and public involvement** Patients and/or the public were not involved in the design, or conduct, or reporting or dissemination plans of this research.

**Patient consent for publication** Not required.

**Ethics approval** This study involves human participants and was approved by Mbarara University of Science and Technology Research Ethics Committee (NO 16/09-19). All the study participants provided written informed consent. Participants gave informed consent to participate in the study before taking part.

**Provenance and peer review** Not commissioned; externally peer reviewed.

**Data availability statement** Data are available upon reasonable request.

**ORCID iD**
Henry Mark Lugobe http://orcid.org/0000-0001-8177-8786

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
