## [Reviewer comments · BMJ Open]

ARTICLE DETAILS

TITLE (PROVISIONAL)	Retinopathy among women with hypertensive disorders of pregnancy attending hospitals in Mbarara city, south-western Uganda: A cross-sectional study
AUTHORS	Zamaladi, Ibrahimu; Ruvuma, Sam; Mceniery, Carmel M.; Kwaga, Teddy; Wilkinson, Ian; Atwine, Daniel; Lugobe, Henry Mark

VERSION 1 – REVIEW

REVIEWER	Annamalai, Radha Sri Ramachandra Institute of Higher Education and Research (Deemed to be University)
REVIEW RETURNED	01-Nov-2022

GENERAL COMMENTS	Please clarify if any of the study participants had associated anterior segment features along with hypertensive retinopathy
--

REVIEWER	Cugati, Sudha The University of Adelaide, Department of Ophthalmology
REVIEW RETURNED	04-Nov-2022

GENERAL COMMENTS	1. Unaided visual acuity is not an ideal indicator of the visual association with hypertensive retinopathy in the study group.2. Coincidentally there was 1 patient with poor vision, 1 with corneal pathology and 1 with non reactive pupil. Is this the same patient? Should this patient be then removed from the analysis? was this a pre existing condition not related to the study?3. Page 11 - line 24 - unable to decipher the limitation4. Since the aim of the study is not to assess the visual acuity in HTN retinopathy amongst pregnant women, it is better not to include the unaided visual acuity in the abstract.
---

REVIEWER	Ayumba, Albert University of California Global Health Institute
REVIEW RETURNED	17-Nov-2022

GENERAL COMMENTS	METHODOLOGY 1. Study participants ; my considered opinion is to exclude those with chronic hypertension if the study aims to inform on the cardiovascular changes in the antenatal period. Otherwise categorise and analyse as per classified hypertensive disease in pregnancy to get the weight of each.2. The time point of the blood pressure measurement should clearly be indicated and i.e wether highest BP recording or at admission or average reading. Important because of its statistical significance for the secondary objective.3. Were fundus picture graded for quality before KW grading?
---

	Also provide specifications of fundus camera in terms of resolution?, portable of table mounted?, it's operator in the study?has it been used before for retinopathy study? RESULTS 1. Did all participants enrolled in the study go through to have the funduscopy? not clear form results section. 2. May consider upload of sample / selected pictures with lessons. ANALYSIS 1. Was it done per patient of per eye. Not clear for results and anaylsis sections DISCUSSION 1. Consider social/healthcare factors which affect pathophysiological pathway in discussing the variance in results with other study. Authors only sited mainly funduscopy technique and sampling time. Discuss why diastolic and not systolic Blood pressure measurements had significance in this study? Limitations 1. consider Hypertensive disease an=s a progressive disease process and therefore one time point assessment in the study design is limited. 2. Criteria for severity involves assessing target organs(aside from BP criteria) which weren't applied in this study.
--	---

VERSION 1 – AUTHOR RESPONSE

Reviewer: 1

Dr. Radha Annamalai, Sri Ramachandra Institute of Higher Education and Research (Deemed to be University)

Comments to the Author:

1. Please clarify if any of the study participants had associated anterior segment features along with hypertensive retinopathy

We have reviewed the original dataset and realized that we had erroneously put 1 participant with a corneal lesion. We have revised our tables.

Reviewer: 2

Dr. Sudha Cugati, The University of Adelaide

Comments to the Author:

1. Unaided visual acuity is not an ideal indicator of the visual association with hypertensive retinopathy in the study group.

We have revised the manuscript and deleted the statement on visual acuity from the abstract.

2. Coincidentally there was 1 patient with poor vision, 1 with corneal pathology and 1 with non reactive pupil. Is this the same patient? Should this patient be then removed from the analysis? was this a pre existing condition not related to the study?

We have reviewed the original dataset and realized that we had erroneously put 1 participant with a corneal lesion. We have revised our tables.

3. Page 11 - line 24 - unable to decipher the limitation

We have re-written the limitation section so that it is clear as shown on page 12.

4. Since the aim of the study is not to assess the visual acuity in HTN retinopathy amongst pregnant women, it is better not to include the unaided visual acuity in the abstract.

We have revised the manuscript and deleted the statement on visual acuity from the abstract.

Reviewer: 3

Dr. Albert Ayumba, University of California Global Health Institute

Comments to the Author:

METHODOLOGY

1. Study participants ; my considered opinion is to exclude those with chronic hypertension if the study aims to inform on the cardiovascular changes in the antenatal period. Otherwise categorise and analyse as per classified hypertensive disease in pregnancy to get the weight of each.

We agree with the reviewer that some of the retinopathy observed in this study may be due to chronic hypertension. However in our study we considered everyone with hypertension without categorising between chronic hypertension and pregnancy induced hypertension. We have included this as part of the limitations of our study as shown on page 12.

2. The time point of the blood pressure measurement should clearly be indicated and i.e whether highest BP recording or at admission or average reading. Important because of its statistical significance for the secondary objective.

We took the blood pressure reading at admission. We did not have multiple blood pressure readings. We have clarified this in the methods section. "The blood pressure was measured at admission or during the antenatal clinic visit." This is on page 5.

3. Were fundus picture graded for quality before KW grading?

Also provide specifications of fundus camera in terms of resolution?, portable of table mounted?, it's operator in the study?has it been used before for retinopathy study?

In our study, a retinal specialist together with trained fundus image graders did the grading before KW grading. We used a portable non-mydratic fundus cameras (Forus 3nethra non-mydratic, Forus Health Pvt Ltd, Bengaluru, India). This has been clarified on page 6. This camera has been used before to study retinopathy in this setting "Arunga, S., Tran, T., Tusingwire, P., Kwaga, T., Kanji, R., Kageni, R., Hortense, L.N., Twinamasiko, A., Kakuhikire, B., Kataate, B. and Kilberg, K., 2022. Diabetic retinopathy screening program in Southwestern Uganda. The Journal of Ophthalmology of Eastern, Central and Southern Africa."

RESULTS

Did all participants enrolled in the study go through to have the fundoscopy? not clear form results section.

All the study participants had fundoscopy done.

2. May consider upload of sample / selected pictures with lesions.

We have uploaded some pictures showing lesions on the retina.

ANALYSIS

1. Was it done per patient of per eye. Not clear for results and analysis sections

We did the analysis per patient and not per eye.

DISCUSSION

1. Consider social/healthcare factors which affect pathophysiological pathway in discussing the variance in results with other study. Authors only sited mainly funduscopy technique and sampling time.

Discuss why diastolic and not systolic Blood pressure measurements had significance in this study?

We reviewed our analysis and re-categorised hypertension into severe and none severe. The results show that severe hypertension was significantly associated with retinopathy (aOR=2.8; 95%CI: 1.36, 5.68) on page 10.

Limitations

1. consider Hypertensive disease as a progressive disease process and therefore one time point assessment in the study design is limited.

We have acknowledged this in our limitation that having blood pressure only measured once could have missed out on participants who might have developed on subsequent visits. We also note that since we did not look out for participants who had chronic hypertension prior to pregnancy, we cannot conclude that the retinopathy observed was solely due to pregnancy hypertension as shown on page 12.

2. Criteria for severity involves assessing target organs(aside from BP criteria) which weren't applied in this study.

We have acknowledged in the limitation that we were not able to do any biochemical tests and therefore we were unable classify participants with severe preeclampsia and whether women with severe preeclampsia were at risk of retinopathy as shown on page 12.